# Site- and enantioselective cross-coupling of saturated *N*-heterocycles with carboxylic acids by cooperative Ni/photoredox catalysis

Xiaomin Shu [1,2], De Zhong[1,2], Qian Huang[1], Leitao Huan[1] & Haohua Huo [1] ✉

Site- and enantioselective cross-coupling of saturated *N*-heterocycles and carboxylic acids–two of the most abundant and versatile functionalities–to form pharmaceutically relevant α-acylated amine derivatives remains a major challenge in organic synthesis. Here, we report a general strategy for the highly site- and enantioselective α-acylation of saturated *N*-heterocycles with in situ-activated carboxylic acids. This modular approach exploits the hydrogen-atom-transfer reactivity of photocatalytically generated chlorine radicals in combination with asymmetric nickel catalysis to selectively functionalize cyclic α-amino C−H bonds in the presence of benzylic, allylic, acyclic α-amino, and α-oxy methylene groups. The mild and scalable protocol requires no organometallic reagents, displays excellent chemo-, site- and enantioselectivity, and is amenable to late-stage diversification, including a modular synthesis of previously inaccessible Taxol derivatives. Mechanistic studies highlight the exceptional versatility of the chiral nickel catalyst in orchestrating (i) catalytic chlorine elimination, (ii) alkyl radical capture, (iii) cross-coupling, and (iv) asymmetric induction.

Chiral α-functionalized azacycles are commonly found in pharmaceutical drugs, natural products, and catalysts for asymmetric synthesis[1–4]. Controlling the stereochemistry of the α-stereocenter is important because it can dramatically alter the biological properties or the asymmetric induction in catalytic reactions. A particularly attractive approach to access enantioenriched α-functionalized azacycles is metal-catalyzed α-functionalization of readily available *N*-heterocycles[5]. Despite being highly desirable, controlling the absolute stereochemistry remains notoriously difficult in C(sp³)−H bond functionalization[6]. Nonetheless, prior studies have shown that α-aryl pyrrolidines can be enantioselectively accessed via (−)-sparteine-mediated asymmetric α-deprotonation followed by cross-coupling with aryl electrophiles[7,8]. In an interesting study, Fu and co-workers demonstrated that enantioenriched α-alkyl pyrrolidines can be achieved through Ni-catalyzed enantioconvergent cross-coupling of racemic α-zincated *N*-Boc-pyrrolidines with alkyl iodides[9,10] (Fig. 1a). Recently, an elegant study by Yu showed that the enantioselective α-arylation of

diverse saturated azacycles can be accomplished via Pd-catalyzed oxidative cross-coupling[11]. Although there have been a limited number of examples for the metal-catalyzed enantioselective α-arylation and α-alkylation, a general strategy for the direct enantioselective α-acylation of common saturated azacycles has yet to be developed[12].

In recent years, the merger of hydrogen-atom-transfer (HAT) photocatalysis with transition metal catalysis has provided a highly enabling platform for C(sp³)−H bond functionalization, wherein traditionally inert C(sp³)−H bonds can be readily transformed into alkyl radicals that subsequently participate in transition metal-catalyzed bond-forming reactions[13–17]. However, controlling the regioselectivity of a HAT process and the enantioselectivity of cross-coupling in such transformations remains an outstanding challenge[18–23]. Each of these two challenges is difficult to overcome individually; solving them simultaneously in a single catalytic system is even more demanding because organic molecules often contain multiple C(sp³)−H bonds with similar bond strengths and steric environments[24–26]. While a

---

[1]State Key Laboratory of Physical Chemistry of Solid Surfaces, Key Laboratory of Chemical Biology of Fujian Province, College of Chemistry and Chemical Engineering, Xiamen University, Xiamen 361005, China. [2]These authors contributed equally: Xiaomin Shu, De Zhong. ✉e-mail: hhuo@xmu.edu.cn

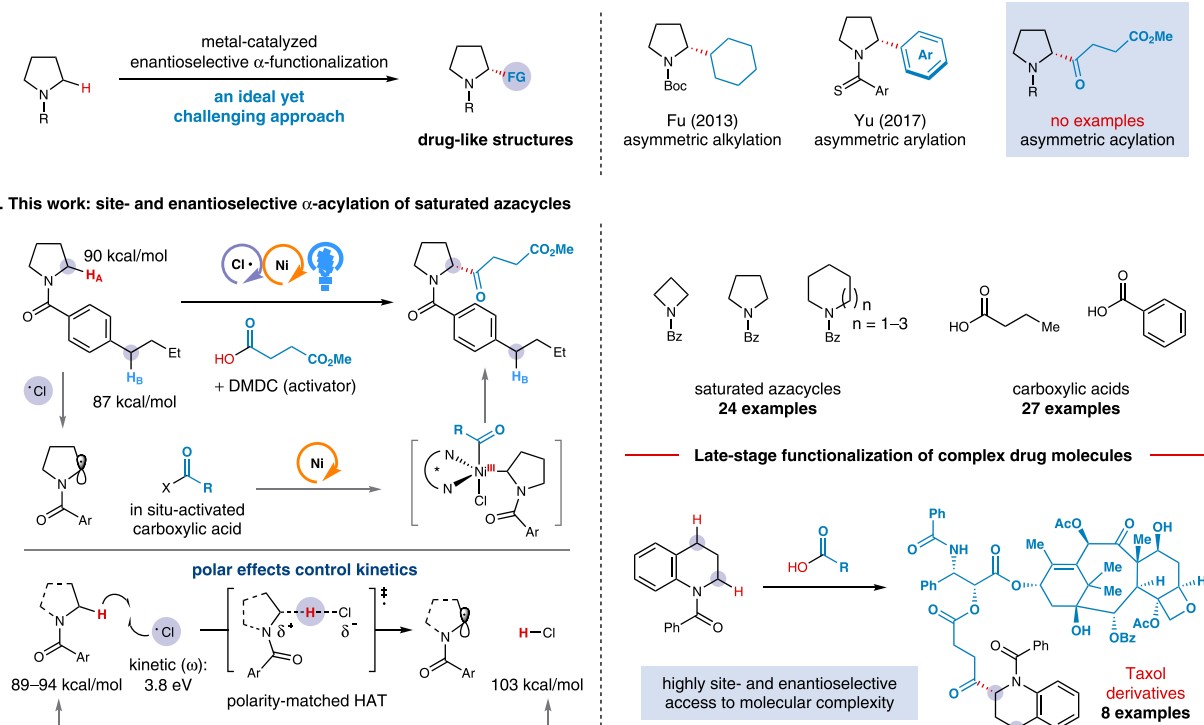

**Fig. 1 | Approaches for transition metal-catalyzed enantioselective α-functionalization of saturated azacycles. a** Prior art for enantioselective α-functionalization of saturated azacycles. **b** This work: site- and enantioselective α-acylation of saturated azacycles. DMDC dimethyl dicarbonate.

variety of metallaphotoredox HAT-mediated bond constructions have begun to achieve levels of regioselectivity[27–34], efficient methods for the enantioselective variants remain underdeveloped[16,22]. In this context, our group recently demonstrated that a photoeliminated chlorine radical (Cl·) can be exploited as the HAT catalyst for an enantioselective α-arylation of saturated azacycles[35]. However, we recognized that, to truly harness the potential of the chlorine-radical-mediated HAT strategy for solving more challenging problems, it must be necessary to conceive of a new catalytic system that should (i) precisely control the site selectivity of HAT among multiple C−H bonds of similar bond strengths and steric environments, (ii) readily render the transformation enantioselective with a simple, chiral base metal catalyst, (iii) advance the repertoire of coupling partners to encompass abundant and readily available functionalities without the need for an exogenous directing group and substrate preactivation steps, and (iv) be amenable to late-stage diversifications with extensive functional group tolerance. With these criteria in mind, we sought to develop a unified approach for the direct site- and enantioselective cross-coupling of carboxylic acids with saturated N-heterocycles—two of the most abundant and versatile functionalities—for the modular synthesis of α-acylated azacycles (Fig. 1b).

We envisioned that an electrophilic Cl·, generated from the photolysis of a high-valent nickel chloride[36–39], can serve as a HAT catalyst to undergo site-selective HAT from cyclic α-amino C−H bonds among multiple C−H bonds of comparable bond strengths. The resulting carbon-centered radical and an in situ-activated carboxylic acid could be subsequently engaged in the nickel-catalyzed cross-coupling in the presence of a common chiral ligand, leading to a Ni(III) species. Finally, reductive elimination would generate the α-acylated product in a site- and enantioselective manner. The main challenge to achieving such a cross-coupling is ensuring the ability of the nickel catalyst to play multiple roles in orchestrating catalytic chlorine generation, α-amino radical capture, alkyl-acyl cross-coupling, and enantioselective induction.

To simultaneously achieve efficient site- and enantioselectivity in such a unified catalytic system, we have postulated three design elements to synergistically maximize the interplay of thermodynamic (enthalpic) and kinetic (polar) effects[40]: (i) the chlorine-radical-mediated HAT process should have a significant thermodynamic driving force due to the formation of a H−Cl bond (BDE: 103 kcal/mol) that is stronger than the most sp³ C−H bonds (α-amino C−H bonds: BDE = 89−94 kcal/mol)[41]; (ii) by taking advantage of polarity matching effect[42,43], the strong electrophilic character of photoeliminated chlorine radicals (local electrophilicity of Cl·: 3.8 eV) should enable a kinetically selective HAT from the most hydridic C−H bonds in the presence of weaker C−H bonds that are less hydridic or polarity-mismatched[44]; and (iii) the recognized ability of nickel catalysts to efficiently bind and stabilize alkyl radicals should render the cross-coupling of alkyl radicals with in situ-activated carboxylic acids enantioselective in the presence of an appropriate chiral ligand[45].

In this work, we report the first general strategy for the direct site- and enantioselective α-acylation of common saturated N-heterocycles with abundant carboxylic acids. This protocol exhibits high chemo-, site-, and enantioselectivity and is applicable to late-stage diversification, including a modular synthesis of previously inaccessible Taxol derivatives.

## Results
### Reaction optimization
To explore both the site- and enantioselectivity for the proposed transformation, a drug-like N-acyl-pyrrolidine (**1**) containing a potentially competitive benzylic C−H bond was selected as the C−H donor substrate (Table 1). Following an extensive evaluation of reaction parameters, we were pleased to find that a common carboxylic acid (**2**) could be readily activated in situ by dimethyl dicarbonate (DMDC), followed by cross-coupling with the α-amino C(sp³)−H bond in the presence of a chiral nickel/bis(oxazoline) catalyst and a common

## Table 1 | Effect of reaction parameters[a]

Reaction scheme:

**1** from retinoic acid receptor agonist + **2**

10 mol% NiCl$_2$·glyme
13 mol% (S, R)-**L1**
1 mol% Ir[dF(CF$_3$)ppy]$_2$(dtbbpy)PF$_6$
1.5 equiv DMDC, 1.5 equiv 2,6-lutidine
benzene, 20 °C, 24 h
427 nm blue LEDs (40 W)
"standard conditions"

→ **3a** + **3b**

| Entry | Variation from standard conditions | Yield (%)[b] | r.r. (3a/3b)[b] | ee of 3a (%)[c] | ee of 3b (%)[c] |
|---|---|---|---|---|---|
| 1 | None | 69 | 28:1 | 95 | – |
| 2 | NiBr$_2$·glyme, instead of NiCl$_2$·glyme | 19 | 1:1.7 | 92 | 73 |
| 3 | NaHCO$_3$, instead of 2,6-lutidine | 58 | 24:1 | 85 | – |
| 4 | i-PrOAc[d], instead of benzene | 71 | 20:1 | 89 | – |
| 5 | 1.5 equiv of NaBr added | 39 | 2.8:1 | 95 | 74 |
| 6 | 1.0 equiv of H$_2$O added | 69 | 39:1 | 93 | – |
| 7 | under air in a capped 4-mL vial | 70 | 27:1 | 95 | – |
| 8 | No Ni, Ir, or light | 0 | – | – | – |

Ligands:

(S, R)-**L1**

(S, R)-**L2**
69% yield, 22:1 r.r.
83% ee (**3a**), 91% ee (**3b**)

(S, R)-**L3**
74% yield, 15:1 r.r.
77% ee (**3a**), 93% ee (**3b**)

(S)-**L4**
77% yield, 18:1 r.r.
65% ee (**3a**), 57% ee (**3b**)

(S)-**L5**
75% yield, 12:1 r.r.
17% ee (**3a**), −27% ee (**3b**)

(S)-**L6**
54% yield, 15:1 r.r.
66% ee (**3a**), 31% ee (**3b**)

(S)-**L7**
28% yield, 24:1 r.r.
17% ee (**3a**)

**L8**
76% yield, 37:1 r.r.

[a]Standard reaction conditions: C–H nucleophile **1** (0.3 mmol), carboxylic acid **2** (0.1 mmol), NiCl$_2$·glyme (10 mol%), (S, R)-**L1** (13 mol%), Ir photocatalyst (1 mol%), DMDC (1.5 equiv), 2,6-lutidine (1.5 equiv), benzene (0.1 M), 427 nm blue LEDs (40 W), 20 °C, 24 h.
[b]The combined yields of **3a** and **3b**, and the regioselective ratios were determined via GC analysis.
[c]Determined via HPLC analysis.
[d]Isopropyl acetate.

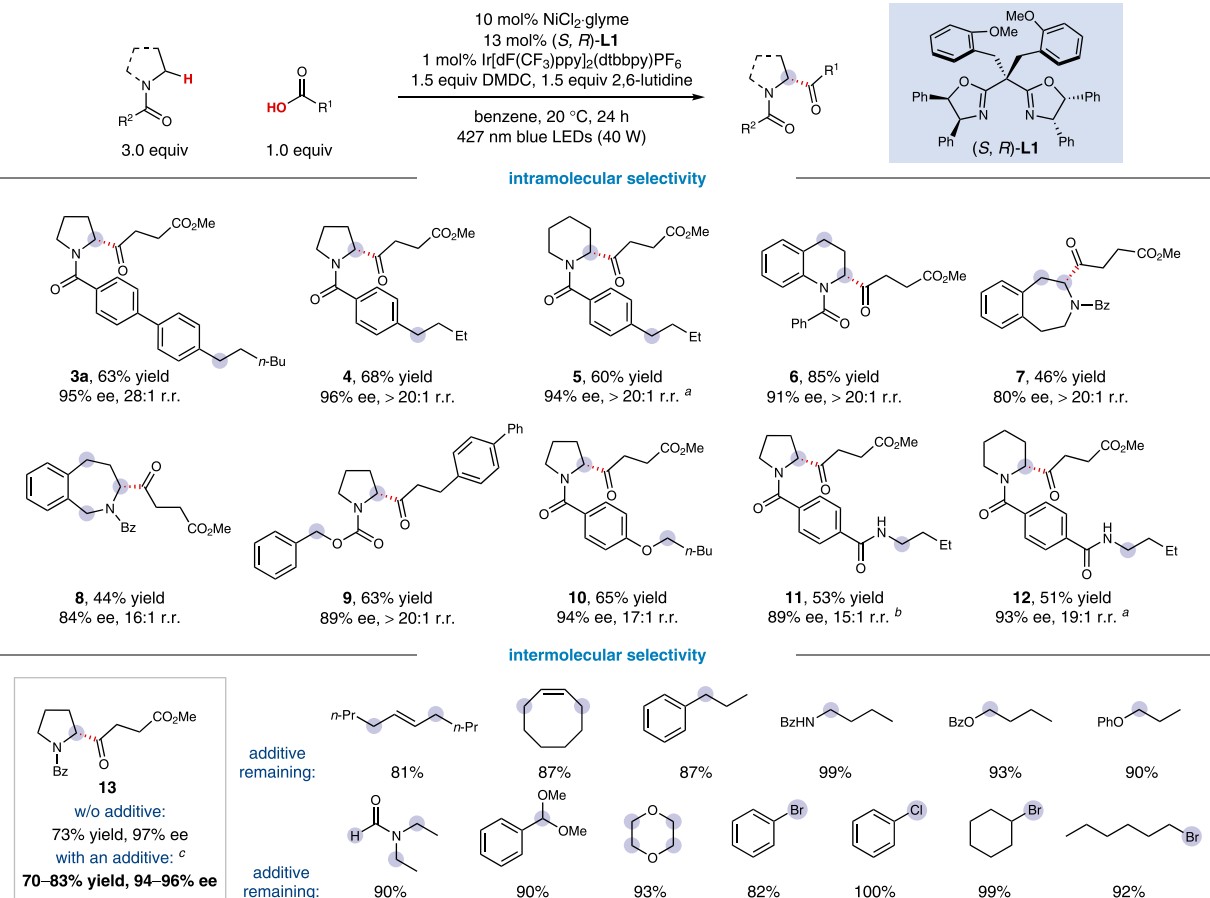

**Fig. 2 | Generality of intramolecular and intermolecular site- and chemos-electivity.** All data represent the average of two experiments, and all yields for the catalytic products are of isolated products. Unless otherwise stated, reactions were conducted on a 0.2 mmol scale under standard conditions. [a](S, R)-**L2** (13 mol%) was used. [b]Without 2,6-lutidine. [c]In the presence of 1.0 equiv of the additive.

photocatalyst Ir[dF(CF₃)ppy]₂(dtbbpy)PF₆ under blue-light irradiation to afford the desired acylation product in 69% yield and with high enantioselectivity and regioselectivity (95% ee, 28/1 r.r., entry 1). This protocol requires no exogenous HAT catalysts and chloride sources, which could mitigate potential concerns about the compatibility of an external HAT catalyst and thus could facilitate the reaction optimization. Attempts to identify an external HAT catalyst for this reaction has been failed (Supplementary Table 6).

In this system, NiCl₂·DME served as both a nickel catalyst precursor and a source of Cl·. Intriguingly, the bromine radical (Br·) relevant conditions[28,46–52] by using NiBr₂·DME as a nickel precatalyst (entry 2) or in the presence of an additional bromide (entry 5) led to unselective formation of both the α-amino and benzylic C–H acylation products in significantly lower yields. Chlorine-relevant conditions provided remarkably better reactivity and selectivity, likely benefiting from a thermodynamically facile (BDEs of H–Cl and H–Br: 103 kcal/mol vs 88 kcal/mol) and polarity-matched HAT process[40]. We attributed this regioselective difference to the better polarity matching of an α-amino C–H bond with Cl· than with Br· (local electrophilicity: 3.8 eV vs 3.6 eV)[44]. It has been shown that a tertiary amide bond of restricted rotation can be locked in a conformation to maximize the interaction between the oxygen lone pair and proximal C–H bonds[53], rendering the cyclic α-amino C–H bond more hydridic and more susceptible to the chlorine-radical-mediated HAT. Empirical evaluation of various ligand scaffolds demonstrated the importance of this newly synthesized ligand (**L1**) for the enantioselective induction. Notably, the replacement of the chiral ligand **L1** with an achiral bipyridine ligand (**L8**) could also generate the product in

good yield and with excellent site selectivity, implying that the site selectivity was not governed by the ligand coordination sphere. Moreover, replacing 2,6-lutiding with NaHCO₃ led to comparable results (entry 3). Isopropyl acetate was also a suitable solvent (entry 4). Importantly, this protocol is not highly sensitive to moisture and air (entries 6 and 7), which could benefit the reaction practicality. Finally, control experiments revealed that the nickel catalyst, photocatalyst, and visible light are indispensable for the acylation product formation (entry 8).

### Synthetic scope
Having established optimal conditions for this site- and enantioselective α-acylation of *N*-heterocycles, we set out to examine the generality of this newly developed acylation protocol. To further probe the generality of the site-selectivity, a range of saturated azacycles containing potentially competitive C(sp³)–H bonds were subjected to this cross-coupling protocol (Fig. 2). To our delight, the α-C(sp³)–H bond of *N*-heterocycles can be selectively acylated in the presence of an acyclic (**3a**, **4**, and **5**) or cyclic (**6–8**) benzylic C–H bond[54–57], an α-oxy C–H bond (**9** and **10**), or an acyclic α-amino C–H bond (**11** and **12**)[47], to afford the coupling products in good yields and with high site- and enantioselectivity (**3a–12**, 44–85% yield, 80–96% ee, ≥15:1 r.r.). In these examples, no isolable regioisomers were observed, and the remaining mass balance basically consisted of methyl esterification product. Notably, this site-selectivity is orthogonal to the previously established HAT-mediated enantioselective C(sp³)–H reactions that generally functionalize the benzylic[31,46,49,54–56] or acyclic α-amino C–H bonds[47], holding the potential for rapid buildup of molecular complexity via

iterative HAT coupling sequences (vide infra). In addition to intramolecular competition, we have also explored the additive robustness to further interrogate the site- and chemoselectivity[58]. As shown in the bottom of Fig. 2, a wide variety of α-C−H substrates that have been widely used in HAT-mediated reactions[16,26], including internal alkenes, alkylarenes, alcohol derivatives, alkyl amine variants, *N*, *N*-diethyl formamide, and acetals, proved to be well tolerated with minimal effect on the yield or enantioselectivity of the desired product. Furthermore, the aryl and alkyl halides commonly susceptible to transition metal-catalyzed cross-coupling reactions were also tolerated additives[35], further demonstrating the orthogonality of this method for the exploration of new chemical space.

We next examined the substrate scope of both coupling partners. As depicted in Fig. 3, a wide variety of structurally diverse saturated azacycles can be readily acylated in good yields and with excellent stereoselectivity (**14−26**, 52–84% yield, 80–97% ee or ≥99:1 dr). For example, the *N*-acyl moiety tolerated electron-rich and electron-deficient substituents (**14** and **15**), a heteroaryl unit (**16**), and carbamates (**21**, also see **9**). This native *N*-acyl unit was strategically used to not only preclude an unproductive *N*-acylation reaction but also a deleterious, photocatalytic amine oxidation. Importantly, this moiety also served as a native functional handle to modulate the hydricity of α-C(sp³)−H bonds[43]. In addition to pyrrolidines, other azacycles of various ring size, including azetidine, piperidine, azepane, azocane, and a lactam, were readily acylated in high efficiency (**17–20** and **22**). Furthermore, a range of 2-subsituted pyrrolidines readily underwent acylation at the 5-methylene position to afford 1,5-bisfunctionalized products with high stereoselectivity (**23−26**). With respect to the electrophile scope, a broad range of electronically and sterically diverse alkyl and aromatic carboxylic acids efficiently coupled with *N*-Bz-pyrrolidine to furnish the α-acylated products in high yields and with excellent enantioselectivity (**27−41**, 46–83% yield, 83–98% ee). Notably, alkyl halides (**33** and **34**) and an aryl chloride (**41**) were well tolerated, providing additional handles for further functionalization. Although carboxylic acids are arguably the most operationally convenient and commercially abundant acyl surrogates, moisture-sensitive carboxylic acid chlorides and anhydrides were also found to be effective coupling partners other than in situ-activated carboxylic acids (Supplementary Table 7)[59–61]. This flexibility holds a potential to fulfill a demand in a specific synthetic scenario.

We next sought to illustrate the robustness of this coupling reaction in the context of drugs, natural products, and biomolecules (the bottom of Fig. 3). Given the extensive representation in bioactive compounds and commercial sources, carboxylic acids are promising vectors for the synthesis of medicinal chemical libraries. Towards this goal, various complex carboxylic acids were found to readily undergo cross-coupling with *N*-Bz-pyrrolidine to generate the α-acylation products in moderate to good yields and with excellent stereoselectivity (**42−62**). For example, pyrrolidine-containing derivatives of oxaprozin (**42**), stearic acid (**43**), *L*-menthol (**46** and **47**), dehydrocholic acid (**48** and **49**), diacetonefructose (**50** and **51**), estrone (**52** and **53**), and (*R*)-citronellic acid (**56** and **57**) were well incorporated in this acylation protocol. These results demonstrate the capacity of our protocol to rapidly access drug-like structural complexity from abundant or readily available starting materials[62].

## Synthetic utility

We next explored the synthetic utility for the late-stage diversification of complex drug-like molecules. To this end, the paclitaxel succinate was selected to couple with four different *N*-heterocycles under slightly modified conditions (Fig. 4a). We were delighted to find that eight Taxol derivatives (**63−70**) can be readily accessed in generally high yields and with excellent stereoselectivities. These results underscore the potential of our method to expedite discovery of drug candidates from existing medicinal chemical libraries without the need

for de novo synthesis[63]. A common limitation of photocatalytic transformations is the scalability often due to the sensitivity of light intensity[64]. The scalability of this method was demonstrated by a 40-fold scale-up in batch for the regioselective synthesis of acylated product **5**. As depicted in Fig. 4b, this gram-scale reaction led to no changes in efficiency. Notably, no specialized light source, glassware, or equipment was required, and the transformation was accomplished in the same timeframe as its small-scale reactions. As a further demonstration of the utility, we sought to rapidly access molecular complexity and functional modularity via iterative coupling sequences by the combination of chlorine- and bromine-mediated enantioselective C(sp³)−H functionalization. To our delight, subjecting the compound **5** to couple with an aryl bromide or an in situ-activated carboxylic acid under our previously developed relevant conditions for bromine-radical-mediated C−H functionalization[35,49], resulted in the formation of the benzylic arylated and acylated products in good yields and with high stereoselectivity (**71−74**). These results further highlight the chemo-, regio-, and enantioselectivity of this newly developed chlorine-mediated C(sp³)−H acylation reaction, as well as its orthogonality to previously established methods[26].

## Mechanistic investigations

We next investigated the reaction mechanism. The control experiments of halide source were first performed to probe for the crucial role of the chloride (Fig. 5a). Accordingly, bromide relevant conditions afforded a significantly lower yield likely due to the lack of a thermodynamic driving force for Br· mediated C−H bond cleavage (Fig. 5a, entry 2). The use of a halide-free Ni(0) source, Ni(COD)₂, even in the presence of additional chloride salts, led to almost no product formation, suggesting that a Ni(0) complex was likely not involved in the catalytic cycle (Fig. 5a, entries 3 and 4)[65]. In contrast, the presence of exogenous chloride salts was found to significantly improve the efficiency for the reaction that uses Ni(acac)₂ as the nickel donor (Fig. 5a, entries 5 and 6). A small amount of product was observed under halide-free conditions (Fig. 5a, entry 5), likely formed by the acac radical-mediated C−H acylation[66]. Next, conducting a competition reaction in the presence of an electron-deficient olefin resulted in the formation of a racemic adduct (**75**), supporting the intermediacy of an α-amino radical in the transformation (Fig. 5b).

To probe for the potential intermediacy of a Ni(II) acyl chloride species, stoichiometric studies using an independently generated Ni(II) acyl chloride were performed. As shown in Fig. 5c, the coupling of *N*-Bz-pyrrolidine with stoichiometric Ni(II) acyl chloride **Ia** under photoredox conditions afforded the product **36** in significantly lower efficiency compared to the catalytic reaction (83% yield, 96% ee). Additionally, nonphotochemical oxidation of this Ni(II) acyl chloride complex to a transient Ni(III) species by [TBPA]SbCl₆ ($E = 1.16$ V vs SCE in CH₂Cl₂)[67], followed by blue-light irradiation did not afford any coupling product **36** (Supplementary Fig. 3)[27]. These data did not support an intermediacy of Ni(II) acyl chloride in the main catalytic pathway (Fig. 5c). Furthermore, cyclic voltammogram studies and luminescence quenching experiments indicate that single-electron oxidation of the Ni(II) dichloride species to a Ni(III) complex by the photoexcited photocatalyst is feasible (Fig. 5d). These results also suggest that the potential intermediacy of a reductively generated acyl radical is likely not involved in the reaction[68]. At present, alternative oxidative chlorine generation by direct SET oxidation of dissociated chlorides cannot be ruled out[69].

On the basis of above results and literature precedents[35,70], a possible mechanism is proposed and shown in Fig. 5e. Initially, Ni(II) dichloride **A** undergoes oxidation by the excited photocatalyst to afford Ni(III) trichloride species **B** (for Ni(II) dichloride **E**: $E_{p/2}^{ox} = +1.02$ V vs SCE in CH₃CN, Fig. 5d; $E_{1/2}$[Ir(III*/II)] = +1.21 V vs SCE in CH₃CN). Subsequently, photolysis of this Ni(III) trichloride complex would provide Cl· and regenerate Ni(II) dichloride **A**[36]. Following the polarity matching effect[42],

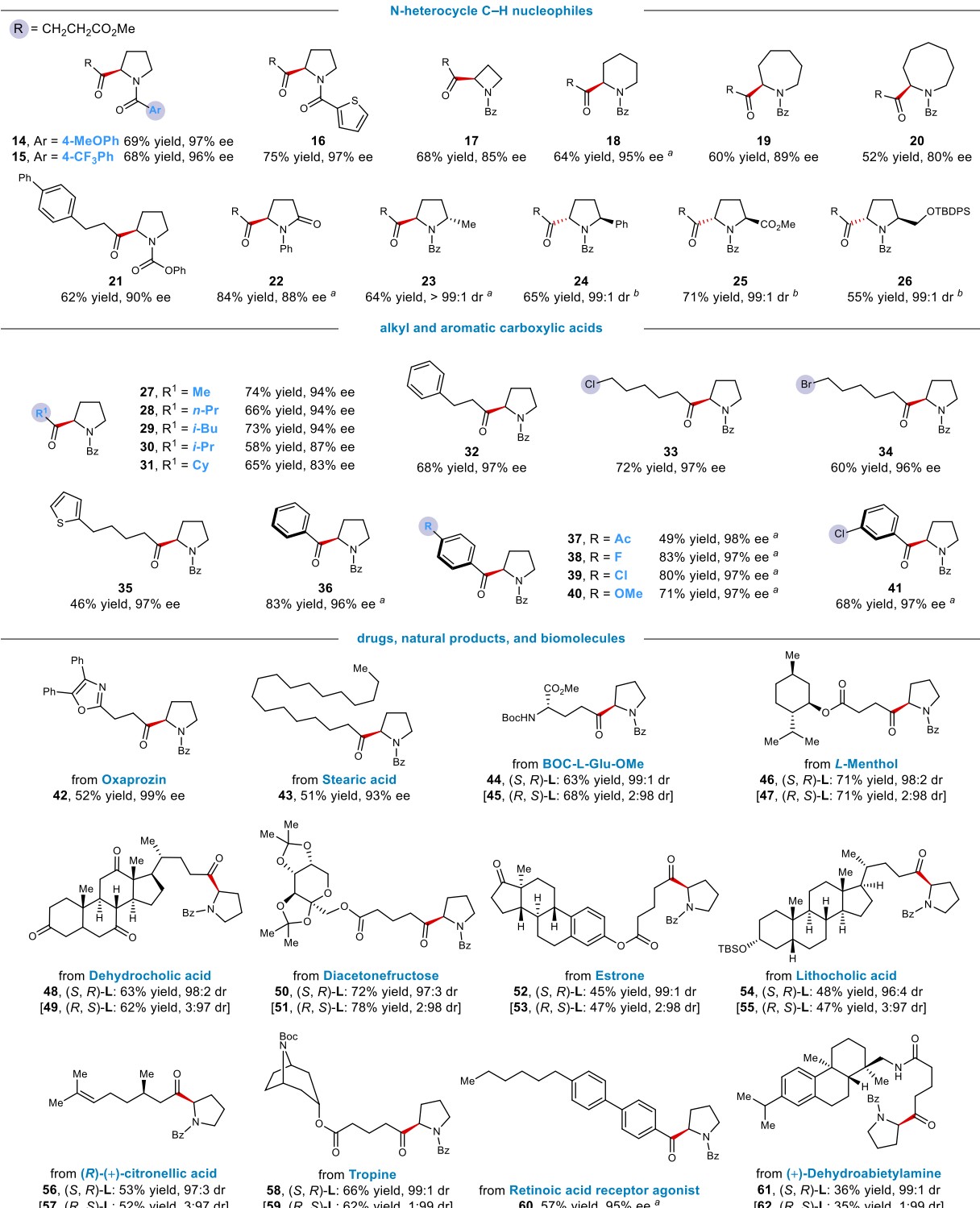

**Fig. 3 | Scope of carboxylic acids and saturated azacycles.** All data represent the average of two experiments, and all yields are of purified products. Unless otherwise stated, reactions were conducted on a 0.2 mmol scale under standard conditions. [a](S, R)-**L2** (13 mol%) was used. [b](R, S)-**L2** (13 mol%) was used.

photoeliminated Cl· would selectively undergo HAT from the most hydridic α-amino C−H bond to produce a prochiral α-amino radical, which would be intercepted by Ni(I) chloride **C**, resulting in the formation of Ni(II) alkyl chloride **D**. A single-electron-transfer (SET) event between Ni(II) complex **D** and the reduced photocatalyst would regenerate ground state photocatalyst and simultaneously afford Ni(I) alkyl complex **E**. The latter one would undergo oxidative addition with an in situ-activated carboxylic

acid to furnish Ni(III) species **F**. Finally, reductive elimination affords the enantioenriched coupling product and regenerates the active Ni(I) chloride catalyst **C** for a new catalytic cycle.

Notably, an alternative mechanism involving chlorine elimination from a Ni(III) acyl chloride[65] or an excited Ni(II) acyl chloride[28] cannot be completely excluded. However, given that (i) stoichiometric reactions did not support the intermediacy of Ni(II) acyl chloride **Ia** in the main catalytic pathway (Fig. 5c), and (ii) a Ni(0) precatalyst led to

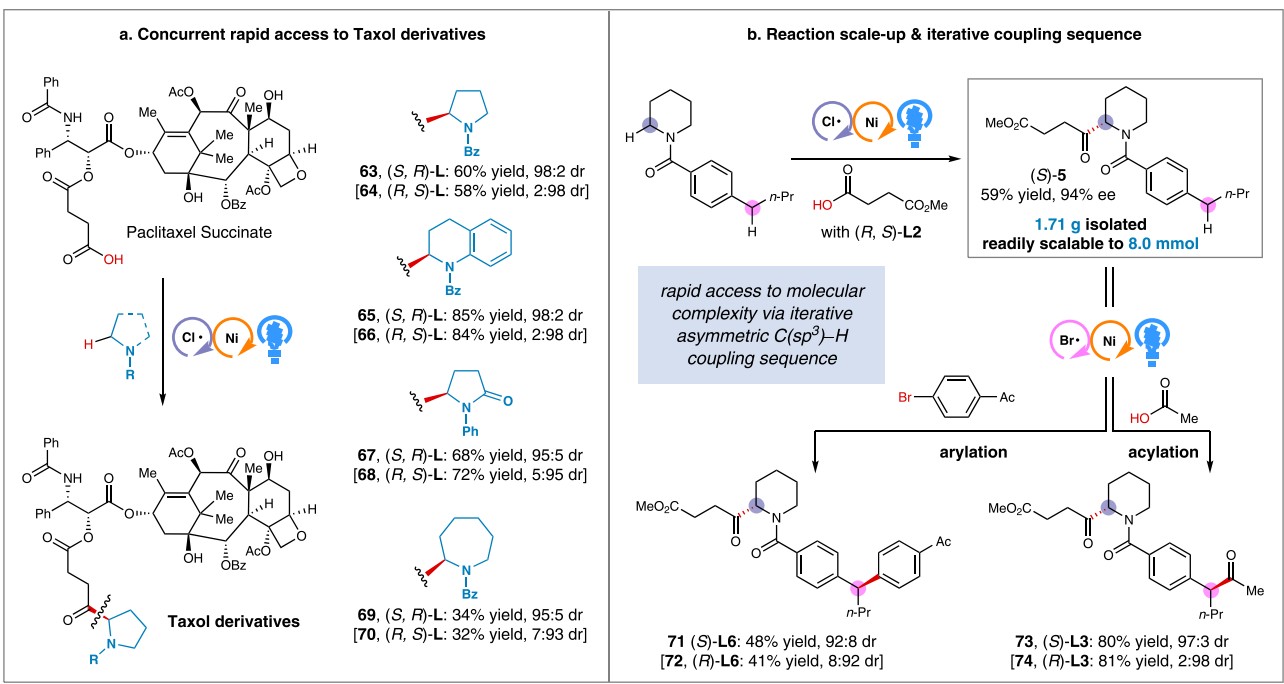

**Fig. 4 | Synthetic utility. a** Concurrent rapid access to Taxol derivative. **b** Reaction scale-up & iterative coupling sequence.

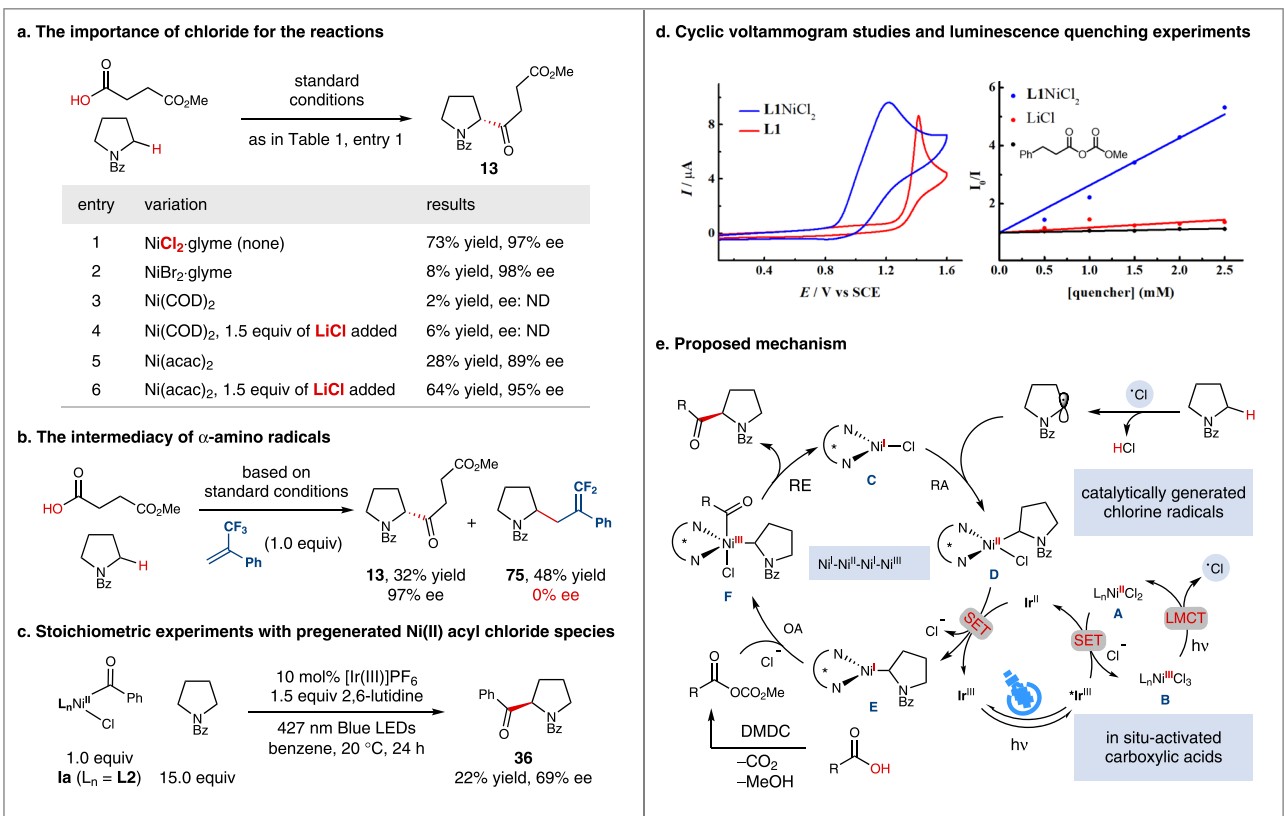

**Fig. 5 | Mechanistic investigations. a** The importance of chloride for the reactions. **b** The intermediacy of α-amino radicals. **c** Stoichiometric experiments. **d** Cyclic voltammogram studies and luminescence quenching experiments. **e** Proposed mechanism. RA radical addition, SET single electron transfer, LMCT ligand-to-metal charge transfer, OA oxidative addition, RE reductive elimination.

almost no product formation (Fig. 5a), the mechanism shown in Fig. 5e must be more operative.

In summary, we have developed a highly site- and enantioselective α-acylation of saturated *N*-heterocycles. This general and modular approach exploits the HAT reactivity of photoeliminated chlorine radicals to selectively functionalize cyclic α-amino C(sp³)−H bonds in the presence of benzylic, allylic, acyclic α-amino, and α-oxy C−H bonds. The robust conditions have enabled direct cross-coupling of

various azacycles with a broad range of abundant carboxylic acids, including natural products and drug-like motifs, in a highly site- and stereoselective fashion. The synthetic utility has been demonstrated by a modular synthesis of eight Taxol derivatives, a facile reaction scale-up, and the applicability in orthogonally iterative C(sp³)–H cross-couplings. We anticipate that this new chlorine-radical-mediated HAT strategy will serve as a general and practical platform for direct enantioselective α-functionalization of saturated azacycles.

## Methods

### General procedure for α-acylation of saturated N-heterocycles with carboxylic acids

In a nitrogen-filled glovebox, Ir[dF(CF$_3$)ppy]$_2$(dtbbpy)PF$_6$ (2.2 mg, 0.002 mmol, 1 mol%), NiCl$_2$·glyme (4.4 mg, 0.020 mmol, 10 mol%), (S, R)-**L1** (18.2 mg, 0.026 mmol, 13 mol%), N-heterocycle (0.60 mmol, 3.0 equiv), a Teflon stir bar, and anhydrous benzene (2.0 mL) was sequentially added to a 15-mL vial. The reaction mixture was stirred at room temperature for 30 min, after which it turned to a pale-yellow suspension (if the carboxylic acid was a solid, it was added as a solid directly at this point). Next, the vial was closed with a PTFE septum cap and wrapped with electrical tape. Then, carboxylic acids (0.20 mmol, 1.0 equiv), 2, 6-lutidine (35.0 μL, 0.30 mmol, 1.5 equiv), and DMDC (32.2 μL, 0.30 mmol, 1.5 equiv) were added sequentially via microsyringe. Next, the vial was transferred out of the glovebox, and then vacuum grease was liberally applied to cover the entire top of the septum cap. Then, the reaction mixture was stirred at 20 °C in an EtOH bath for 1 min before being irradiated with a 40 W the blue LED lamp (Kessil PR160L, 427 nm). The reaction was stirred under blue LED irradiation at 20 °C for 24 h. The reaction mixture was then passed through a short pad of silica gel, with acetone as the eluent (~20 mL). The regioselectivity was determined via GC analysis of the crude reaction mixture. The resulting mixture was concentrated, and the residue was purified by flash chromatography on silica gel or preparative thin-layer chromatography on silica gel to provide the desired product, and the stereoselectivity was determined via HPLC analysis.

## Data availability

The data that support the findings of this study are available within the article and the Supplementary Information. Details about materials and methods, experimental procedures, characterization data, mechanistic studies, NMR and HPLC spectra are available in the Supplementary Information. The crystallographic data for compound **40** can be obtained free of charge from the Cambridge Crystallographic Data Centre (CCDC) under reference number 2206850.

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

## Acknowledgements

The authors are grateful for financial support provided by the National Key R&D Program of China (2021YFA1502500), National Natural Science Foundation of China (22071203), and Fundamental Research Funds for the Central Universities (20720210014).

## Author contributions

X.S. and D.Z. contributed equally to this work. X.S., D.Z., Q.H., and L.H. performed the experiments and analyzed the data. H.H. designed and directed the project and wrote the manuscript.

## Competing interests

The authors declare no competing interests.
