## [Peer Review File · Nature Communications]

REVIEWER COMMENTS

Reviewer #1 (Remarks to the Author):

Photoinduced asymmetric Csp³-H bond functionalization allows the efficient access to value-added chiral molecules in high atom economy and sustainability, but remains a prominent challenge in organic synthesis despite of previous advances. In this manuscript, Huo and co-workers describe the asymmetric α -acylation of saturated azacycles with good site- and enantio-selectivities via the merger of photoredox and Ni catalysis. Taking advantages of photocatalytically generated chlorine radical, various chiral N-heterocycles could be accessed from widely available starting materials, including carboxylic acids and saturated azacycles. Moreover, the newly developed catalytic system exhibits good site-selectivity among various activated Csp³-H bonds via kinetic polar effect control. As a consequence, this method features relatively broad substrate scope with good functional group compatibility, which could find broad application in the late-stage functionalization of complexed molecules in pharmaceuticals. Preliminary mechanistic experiments were conducted to provide insights into the reaction pathway. Overall, this manuscript represents a novel and important study in this filed, especially considering the good regioselectivity control among various Csp³-H bonds and potential synthetic utility. So, I recommend this paper to be published in Nature Communications.

A few comments:

1. Table 1, entry 7, "under air in a capped vial". I think it is better to include the volume of vial to provide more information.

Reviewer #2 (Remarks to the Author):

Huo and coworkers report on a dual photoredox/Ni-catalyzed methodology for the direct enantioselective acylation of α -C(sp³)-H bonds of saturated N-heterocycles from widely available carboxylic acids that are activated in situ. Chiral α -functionalized azacycles represent a very important class of chemicals, their synthesis has thus attracted considerable interest. In this domain the development of catalytic processes allowing the direct enantioselective functionalization of the α -C(sp³)-H bonds of azacycles remains highly desirable but very challenging. To date, the few examples of such reactions are α -alkylations/arylations implying C-H deprotonations followed by Pd or Ni-catalyzed cross-couplings. The only example of acylation of α -C-H bonds functionalization of tertiary amines from aldehydes exploits a dual catalytic system combining a chiral heterocyclic carbene catalyst with a Ru(bpy)₃ photocatalyst. The methodology is limited to the acylation of N-phenyltetrahydroisoquinolines.

The innovative methodology described by Huo and coworkers exploits: 1) chlorine atom to mediate direct HAT from the α -C(sp³)-H bonds. The chlorine atoms are photogenerated thanks to Ir(III)-mediated photoredox catalysis; 2) Carboxylic acids as acylating agents activated in situ by a dicarbonate; 3) A nickel(II) precatalyst LNi(II)Cl₂ generated in situ from NiCl₂ and a chiral bis-oxazoline ligand. The reactions are conducted at 20 °C in benzene under blue light LED irradiation. From this system 4-, 5-, 6-, 7- and 8-membered ring saturated azacycles were acylated enantioselectively. The scope of carboxylic acid that could be employed is broad including alkyl- and aryl-carboxylic acids bearing various functional groups such as halogen atoms or heterocycles. Of particular interest is that the methodology works nicely from carboxylic acid-derived drugs, natural products and biomolecules. Impressive examples of direct enantioselective functionalization of pacitaxel succinate are reported. Importantly, the ee are most often > 95%, the yields moderate to good, and the chemoselectivity strongly in favor of the α -C(sp³)-H bonds of cyclic amines. Finally, preliminary mechanistic studies were conducted highlighting the key role of the chlorine radical and Ni(I) as a catalytic species. A plausible mechanism is proposed on account of the experimental observations.

Overall, considering the quality of the work, the efficiency of the methodology and the novelty/importance of the reaction, I recommend publication in Nat. Comm. I have however some points that should be answered/commented:

1- In the general procedures given in the SI for the catalytic experiments are conducted under

strictly anaerobic conditions. The reaction mixture is prepared in a glovebox, the vial cap is wrapped with electrical tape and vacuum grease is applied to cover the cap entirely before irradiation of the vial conducted outside the glovebox. This contrast with the much more practical conditions reported in entry 7 of table 1 showing that the reaction between 1 and 2 conducted under air in a capped vial affords the same yield, regioselectivity and enantioselectivity. Did the authors try these "aerated" conditions on other reactions? In other words, are there any justifications for employing strictly anaerobic conditions for this methodology? I would suggest to test these conditions to a few other substrates to verify if the non-deaerated conditions lead to the same results. If this is the case, I would comment these results in the text and add a detailed experimental procedure in the SI.

2- Previous work from Barriault (ACIE 2018, 57, 15664) showed that the Ir(III) excited state of Ir[dF(CF₃)ppy]₂(dtbbpy)PF₆, the same photocatalyst than the one employed in this study, can oxidize chloride ions into chlorine radicals. This could thus potentially represent a competitive pathway from the one proposed by the authors which implies a LNi(II)Cl₂/LNi(III)Cl₃ catalytic cycle with the chlorine atoms produced by photolysis of LNi(III)Cl₃ (LMCT). Luminescence quenching experiments presented in Fig. 5 show that the Ir(III) excited state is quenched much more efficiently by LNiCl₂ than LiCl. This supports the generation of LNi(III)Cl₃ from LNi(II)Cl₂/Cl⁻ by SET but is not a direct proof that such SET process occurs. Moreover, the chloride concentration during catalysis could be significantly high, since chlorides will be generated during the activation of the precatalyst LNi(II)Cl₂ (10 mol%) into the proposed catalytically active LNi(I)Cl species (intermediate C). I note that, as usual, this key step is not described/discussed in the mechanism. Overall, it seems to me that a mechanism in which the chlorine radical will be generated by direct SET to the Ir(III) excited state cannot be ruled out at this stage.

3- L172 in the "Synthetic scope" paragraph the yields for products 42-62 are considered to be "generally high yields". In my opinion the yields should be qualified of "moderate to good". For instance, ten of these compounds are obtained with yields lower than 53%.

4- The configuration of the stereocenter of 5 and 71-74 in Fig. 4 should be inverted (acyl chain above the cycle). As described in the SI, the 8 mmole scale synthesis of 5 was conducted with the (S,R)-L2 catalyst which provides the opposite configuration, similarly to the (S,R)-L1 catalyst (see table 1 and fig. 2).

Point-by-Point Response to the Comments

Reviewer 1:

“Photoinduced asymmetric Csp³-H bond functionalization allows the efficient access to value-added chiral molecules in high atom economy and sustainability, but remains a prominent challenge in organic synthesis despite of previous advances. In this manuscript, Huo and co-workers describe the asymmetric α -acylation of saturated azacycles with good site- and enantio-selectivities via the merger of photoredox and Ni catalysis. Taking advantages of photocatalytically generated chlorine radical, various chiral N-heterocycles could be accessed from widely available starting materials, including carboxylic acids and saturated azacycles. Moreover, the newly developed catalytic system exhibits good site-selectivity among various activated Csp³-H bonds via kinetic polar effect control. As a consequence, this method features relatively broad substrate scope with good functional group compatibility, which could find broad application in the late-stage functionalization of complexed molecules in pharmaceuticals. Preliminary mechanistic experiments were conducted to provide insights into the reaction pathway. Overall, this manuscript represents a novel and important study in this filed, especially considering the good regioselectivity control among various Csp³-H bonds and potential synthetic utility. So, I recommend this paper to be published in Nature Communications.”

Response: We thank the reviewer for the very positive comments.

Comment 1: “Table 1, entry 7, “under air in a capped vial”. I think it is better to include the volume of vial to provide more information.”

Response: To clarify this, this information has been added to Table 1 (entry 7).

Reviewer 2:

“Huo and coworkers report on a dual photoredox/Ni-catalyzed methodology for the direct enantioselective acylation of α -C(sp³)-H bonds of saturated N-heterocycles from widely available carboxylic acids that are activated in situ. Chiral α -functionalized azacycles represent a very important class of chemicals, their synthesis has thus attracted considerable interest. In this domain the development of catalytic processes allowing the direct enantioselective functionalization of the α -C(sp³)-H bonds of azacycles remains highly desirable but very challenging. To date, the few examples of such reactions are α -alkylations/arylations implying C-H deprotonations followed by Pd or Ni-catalyzed cross-couplings. The only example of acylation of α -C-H bonds functionalization of tertiary amines from aldehydes exploits a dual catalytic system combining a chiral heterocyclic carbene catalyst with a Ru(bpy)₃ photocatalyst. The methodology is limited to the acylation of N-phenyltetrahydroisoquinolines.

The innovative methodology described by Huo and coworkers exploits: 1) chlorine atom to mediate direct HAT from the α -C(sp³)-H bonds. The chlorine atoms are photogenerated thanks to Ir(III)-mediated photoredox catalysis; 2) Carboxylic acids as acylating agents activated in situ by a dicarbonate; 3) A nickel(II) precatalyst LNi(II)Cl₂ generated in situ from NiCl₂ and a chiral bis-oxazoline ligand. The reactions are conducted at 20 °C in benzene under blue light LED irradiation. From this system 4-, 5-, 6-, 7- and 8-membered ring saturated azacycles were acylated

enantioselectively. The scope of carboxylic acid that could be employed is broad including alkyl- and aryl-carboxylic acids bearing various functional groups such as halogen atoms or heterocycles. Of particular interest is that the methodology works nicely from carboxylic acid-derived drugs, natural products and biomolecules. Impressive examples of direct enantioselective functionalization of pacitaxel succinate are reported. Importantly, the ee are most often > 95%, the yields moderate to good, and the chemoselectivity strongly in favor of the α -C(sp³)-H bonds of cyclic amines. Finally, preliminary mechanistic studies were conducted highlighting the key role of the chlorine radical and Ni(I) as a catalytic species. A plausible mechanism is proposed on account of the experimental observations.

Overall, considering the quality of the work, the efficiency of the methodology and the novelty/importance of the reaction, I recommend publication in Nat. Commn.”

Response: We thank the reviewer for the very positive comments.

Comment 1: “In the general procedures given in the SI for the catalytic experiments are conducted under strictly anaerobic conditions. The reaction mixture is prepared in a glovebox, the vial cap is wrapped with electrical tape and vacuum grease is applied to cover the cap entirely before irradiation of the vial conducted outside the glovebox. This contrast with the much more practical conditions reported in entry 7 of table 1 showing that the reaction between 1 and 2 conducted under air in a capped vial affords the same yield, regioselectivity and enantioselectivity. Did the authors try these “aerated” conditions on other reactions? In other words, are there any justifications for employing strictly anaerobic conditions for this methodology? I would suggest to test these conditions to a few other substrates to verify if the non-dearated conditions lead to the same results. If this is the case, I would comment these results in the text and add a detailed experimental procedure in the SI.”

Response: The experiment under air in a capped vial is provided in order to establish that the reaction is not *highly*-sensitive to trace amounts of oxygen. To make this clear, we have conducted comparison experiments for the formation of three structurally diverse products other than the product **3a** depicted in Table 1. As shown below, the presence of air (in a capped 4-mL vial) led to almost identical yields but slightly lower ee's. To ensure the reproducibility and the best catalytic performance, we recommend that the reaction be set up using a glovebox as described in Section 1.3 or 1.4 of the SI. We have added these results to Section 1.3 of SI and also added a note to Section 1.3 to clarify this:

“Note: Although the reaction is not highly sensitive to oxygen and can be performed under air in a capped 4-mL vial (as show below), to ensure the reproducible results, the coupling is recommended to set up using a glovebox, as described above.”

Comment 2: “Previous work from Barriault (ACIE 2018, 57, 15664) showed that the Ir(III) excited state of Ir[dF(CF₃)ppy]₂(dtbbpy)PF₆, the same photocatalyst than the one employed in this study, can oxidize chloride ions into chlorine radicals. This could thus potentially represent a competitive pathway from the one proposed by the authors which implies a LNi(II)Cl₂/LNi(III)Cl₃ catalytic cycle with the chlorine atoms produced by photolysis of LNi(III)Cl₃ (LMCT). Luminescence quenching experiments presented in Fig. 5 show that the Ir(III) excited state is quenched much more efficiently by LNiCl₂ than LiCl. This supports the generation of LNi(III)Cl₃ from LNi(II)Cl₂/Cl⁻ by SET but is not a direct proof that such SET process occurs. Moreover, the chloride concentration during catalysis could be significantly high, since chlorides will be generated during the activation of the precatalyst LNi(II)Cl₂ (10 mol%) into the proposed catalytically active LNi(I)Cl species (intermediate C). I note that, as usual, this key step is not described/discussed in the mechanism. Overall, it seems to me that a mechanism in which the chlorine radical will be generated by direct SET to the Ir(III) excited state cannot be ruled out at this stage.”

Response: To clarify this, we have cited above mentioned literature and added a note to the Section of Mechanistic Investigations in the text:

“At present, alternative oxidative chlorine generation by direct SET oxidation of dissociated chlorides cannot be ruled out.”

Comment 3: “L172 in the “Synthetic scope” paragraph the yields for products 42-62 are considered to be “generally high yields”. In my opinion the yields should be qualified of “moderate to good”. For instance, ten of these compounds are obtained with yields lower than 53%.”

Response: To clarify this, we have changed “generally high yields” to “moderate to good yields”.

Comment 4: “The configuration of the stereocenter of **5** and **71-74** in Fig. 4 should be inverted (acyl chain above the cycle). As described in the SI, the 8 mmole scale synthesis of **5** was conducted with the (S, R)-**L2** catalyst which provides the opposite configuration, similarly to the (S, R)-**L1** catalyst (see table 1 and fig. 2).”

Response: This typo has been corrected in Fig. 4 and the SI. The 8.0 mmol reaction was performed with the (R, S)-**L2** rather than its enantiomer (S, R)-**L2**.

REVIEWERS' COMMENTS

Reviewer #2 (Remarks to the Author):

All comments have been addressed satisfactorily by the authors. I thus recommend publication.